# Do WGANs succeed because they minimize the Wasserstein Distance? Lessons from Discrete Generators

**Ariel Elnekave, Yair Weiss**
The Hebrew University of Jerusalem
{Ariel.Elnekave, Yair.Weiss}@mail.huji.ac.il

## Abstract

Since WGANs were first introduced, there has been considerable debate on whether their success in generating realistic images can be attributed to minimizing the Wasserstein distance between the distribution of generated images and the training distribution. In this paper, we present theoretical and experimental results that show that WGANs *do* minimize the Wasserstein distance but the form of the distance that is minimized depends highly on the discriminator architecture and its inductive biases. Specifically, we show that when the discriminator is convolutional, WGANs minimize the Wasserstein distance between *patches* in the generated images and the training images, not the Wasserstein distance between images. Our results leverage the advantages of *discrete* generators for which the Wasserstein distance between the generator distribution and the training distribution can be computed exactly and the minima can be characterized analytically. We present experimental results with discrete GANs that generate realistic fake images (comparable in quality to their continuous counterparts) and present evidence that they are minimizing the Wasserstein distance between real and fake patches and not the distance between real and fake images. Our code is available at https://github.com/ariel415el/DiscreteGANs.git

## 1 Introduction

In a seminal paper, (Arjovsky et al., 2017) showed the relationship between generative adversarial networks (GANs) and the Wasserstein distance ($W_1$) between two distributions. They argued that when the data lies on a low dimensional manifold, the Wasserstein distance is a more sensible optimization criterion compared to the KL divergence and showed that the Wasserstein distance can be approximately optimized using an adversarial game between two neural networks: a generator network and a critic network. The key difference between their method, the Wasserstein GAN (WGAN), and previous GANs is that the critic is regularized to be 1-Lipshitz, and a great deal of subsequent research has focused on improved regularization techniques (Gulrajani et al., 2017; Miyato et al., 2018; Anil et al., 2019). WGANs have been used in many applications and can provide excellent sample quality in different challenging image datasets (Radford et al., 2015; Isola et al., 2017; Brock et al., 2018; Karras et al., 2020; Sauer et al., 2022; Pan et al., 2023).

In recent years, however, the connection between the success of GANs and the Wasserstein distance has been questioned (Stanczuk et al., 2021; Fedus et al., 2018; Mallasto et al., 2019; Korotin et al., 2022; Milne & Nachman, 2022). The first criticism is the extent to which WGANs do minimize the Wasserstein distance. Several authors have shown that approximately minimizing $W_1$ using WGANs can yield a poor approximation (Pinetz et al., 2019; Mallasto et al., 2019; Stanczuk et al., 2021; Korotin et al., 2021). A second criticism is whether minimizing the Wasserstein distance between two distributions is a sensible optimization criterion for generative models of images. Figure 1 shows a result from Mallasto et al. (2019): a model that does a much better job of minimizing an empirical estimate of the Wasserstein distance actually produces results of much lower visual quality. This has led to an alternative view whereby "GANs succeed because they fail to approximate the Wasserstein distance" (Stanczuk et al., 2021) and that GANs should not be seen as minimizing a loss function (Goodfellow et al., 2020; Fedus et al., 2017). Many papers have completely abandoned

Gradient Clipping
Poor Batch Wasserstein Distance        c-transform
Good Batch Wasserstein Distance

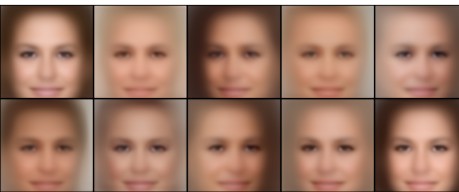

Figure 1: When a WGAN generator is trained using gradient clipping, the approximation to Wasserstein distance is poor and a batch of generated faces have a higher Wasserstein distance to a batch from the training set compared to when the "c-transform" method is used (Mallasto et al., 2019). Nevertheless, the visual quality is higher with the poor approximation.

the distribution- matching approach and focused instead on analyzing the adversarial game and its equilibrium (Sidheekh et al., 2021; Farnia & Ozdaglar, 2020; Schäfer et al., 2019; Qin et al., 2020).

A seemingly simple approach to answering the question of whether WGANs succeed because they minimize $W_1$ is to measure the Wasserstein distance between $P_\theta$, the distribution over fake images defined by the GAN, and $P_{data}$, the true distribution, and to compare that distance with alternative methods that minimize $W_1(P, P_{data})$. Unfortunately, an exact calculation of $W_1(P, Q)$ where $P, Q$ are continuous distributions can only be done for a limited class of distributions. In previous works (e.g. figure 1), an empirical approximation to $W_1(P, Q)$ was used, but this approximation is known to be poor for high-dimensional data such as images (Weed & Bach, 2019). Thus as long as we use continuous, non-parametric distributions for the data and the generated images, it is impossible to give a rigorous answer to whether WGANs minimize $W_1$.

In this paper, we present an alternative approach that allows us to give a rigorous answer. We leverage the advantages of *discrete* GANS. These GANS are identical to the standard GANs in which a noise vector $z$ is passed through a neural network $f_\theta$ to generate an image. But in discrete GANs, the noise vector $z$ is sampled uniformly from $M$ possible fixed noise vectors and thus the generator can generate at most $M$ possible images. Our work was motivated by our initial findings that when $M$ is sufficiently large, discrete GANs generate images that are of comparable quality to that of standard GANs with the same architecture. Figure 2 shows images generated by a variant of FastGAN [1] that we trained as a discrete GAN with $M = 70,000$ fixed noise vectors. The results are comparable in quality to training the same architecture with continuous noise vectors and similar results are obtained with other values of $M$.

By using a discrete GAN we obtain the following advantages:

- We can exactly compute the Wasserstein distance between the GAN distribution $P_\theta$ and the empirical distribution $P_{data}$

- We can analytically characterize the optimal distribution $P_\theta$ that minimizes $W_1(P_\theta, P_{data})$ for different values of $M$.

- We can directly optimize $W_1(P_\theta, P_{data})$ and compare these (locally) optimal solutions to the ones found by WGANs.

In this paper, we leverage these advantages of discrete GANs to provide theoretical and experimental evidence that successful WGANs *do* minimize the Wasserstein distance but the form of the distance that is minimized depends highly on the discriminator architecture and its inductive biases. Specifically, we show that when the discriminator is convolutional, WGANs minimize the Wasserstein distance between *patches* in the generated images and the training images, not the Wasserstein distance between images.

---

[1]See appendix B.2

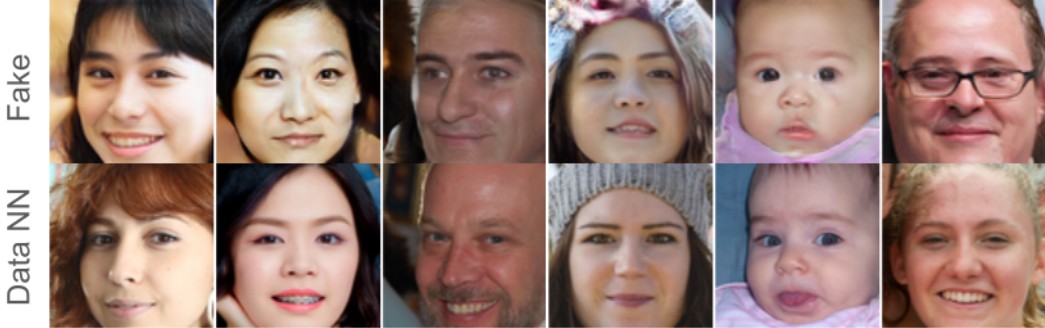

Figure 2: (top:) Images generated by a discrete version of FastGAN trained on 128x128 images from FFHQ with $M = 70,000$. Even though the generator is discrete the visual quality is high and comparable to that of the continuous generator. Similar results are obtained with different values of $M$. (botttom:) the closest training example to each generated image.

## 2 EXACT COMPUTATION OF $W_1$ IN DISCRETE SETTING AND CHARACTERIZATION OF THE MINIMUM.

We start by reviewing the connection between Wasserstein distance and WGANs. The Wasserstein distance $W_1(P, Q)$ between two distributions is defined as:

$$W_1(P, Q) = \inf_{\gamma \in \Pi(P,Q)} E_{x,y \sim \gamma} \|x - y\| \tag{1}$$

where $\Pi(P, Q)$ denotes the set of joint distributions whose marginal probabilities are $P, Q$. The connection to GANs is more evident in the dual form:

$$W_1(P, Q) = \max_{f \in F_1} E_P(f) - E_Q(f) \tag{2}$$

where $F_1$ is the class of 1-Lipschitz bounded functions. Thus if we denote by $P$ the distribution over images defined by the generator and $Q$ the training distribution, the minimization of $W_1(P, Q)$ can be performed using an adversarial game in which the generator attempts to decrease $E_P(f) - E_Q(f)$ and the discriminator, or critic $f$, attempts to increase $E_P(f) - E_Q(f)$.

As mentioned in the introduction, the Wasserstein distance between arbitrary $P, Q$ cannot be computed efficiently and in this paper we leverage the advantages of discrete distributions for which exact computation and optimization is possible.

**Definition 2.1. (Discrete distribution)** Given a set of $N$ points $\{x_i\}_{i=1}^N$ we denote the discrete distribution defined by these points by $P_{\{x_i\}}(x) = \frac{1}{N}\delta(x - x_i)$, where $\delta$ is the Dirac delta function.

**Definition 2.2. (Discrete $W_1$):** Given a set of $M$ points $\{x_i\}$ and a second set of $N$ points $\{y_i\}$ the Wasserstein distance between the discrete distributions defined by the two sets is given by:

$$W_1(P_{\{x\}}, P_{\{y\}}) = \min_{\pi} \sum_{i,j} \pi_{ij} \|y_i - x_j\| \tag{3}$$

with $\pi_{ij}$ a $M \times N$ matrix whose elements are in $[0, 1]$ and satisfies $\sum_i \pi_{ij} = \frac{1}{M}$, $\sum_j \pi_{ij} = \frac{1}{N}$.

Unlike the continuous case, the solution for the optimal transport matrix $\pi$ can be done in polynomial time so that $W_1(P_{\{x\}}, P_{\{y\}})$ can be computed exactly. The connection to discrete GANs is described in the following definition:

**Definition 2.3. (Discrete $W_1$ optimization problem)**. Given a set of of $N$ points $\{y_i\}$ the discrete $W_1$ optimization problem is to find a set of $M$ points $\{x_j\}$ such that $W_1(P_{\{x\}}, P_{\{y\}})$ is minimal.

| $M < N$ | $M = N$ | $M > N$ |
|---|---|---|
| 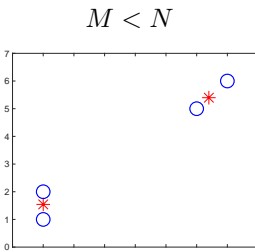 | 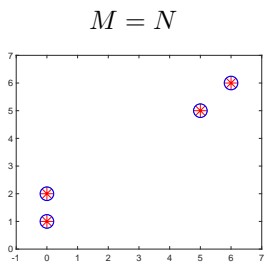 | 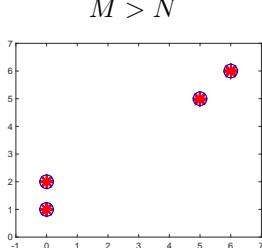 |

Figure 3: In the discrete Wasserstein minimization problem, we approximate a training distribution consisting of $N$ examples (blue circles) with a generated distribution consisting of $M$ examples (red asterisk). In this paper we present an algorithm (OT-means) for solving this problem and we characterize the optimal solution. Specifically, we show that when $M < N$ the optimal solution generates images that are linear combinations of training examples, while for $M = N$ and $M > N$ it copies the training examples.

Figure 3 illustrates this problem. The training training distribution has $N$ examples (blue circles) and we try to approximate it with a generated distribution with $M$ examples (red asterisks). The figure shows numerical solutions to this problem using an iterative algorithm that we call "OT-Means".

**Algorithm (OT-Means)**. Repeat until convergence:

- Given the current estimate of the generated points $\{x_j\}$ set $\pi$ to be the optimal transport matrix betwen $\{x_j\}_{j=1}^M$ and the training set $\{y_i\}_{i=1}^N$

- Given $\pi$ minimize:
$$x_j = \arg \min_x \sum_i \pi_{ij} \|y_i - x\|$$

  This minimization is the geometric median problem and can be performed using iteratively reweighted least squares (Weiszfeld, 1937).

It is easy to show that this algorithm decreases $W_1(P_{\{x\}}, P_{\{y\}})$ at each iteration.

Figure 3 shows the output of OT-means on the same toy dataset with different values of $M$. It can be seen that when $M < N$ the optimal solution generates samples that are linear combinations of training examples, while for $M = N$ and $M > N$ it copies the training examples. The following theorem characterizes the solutions to the problem.

**Theorem 2.4.** *For $M = N$ or $M > N$, $\frac{M}{N} = k$ the optimal solution to the discrete $W_1$ optimization problem is for the generator to copy the training examples. For $M < N$, at any local minimum of the problem, each generated sample is a linear combination of at least $N/M$ training examples.*

*Proof.* (sketch) The result for $M = N$ or $M = kN$ follows from the fact that $W_1 \geq 0$ and copying the examples yields $W_1 = 0$. The result for $M < N$ follows by differentiating $W_1(P_{\{x\}}, P_{\{y\}})$ with respect to a specific $x_j$ and setting the gradient to zero. See appendix A.1 for full proof. $\square$

Given these results, we can now rigorously answer the following question: is the success of the discrete GAN shown in figure 2 due to minimizing $W_1$ ? This GAN was trained with $M = N = 70,000$ thus the optimal solution is to simply copy the training examples. But as can be seen, in figure 2, the discrete GAN *is not* copying the training images and when we compute the exact $W_1$ for this problem we see that is far from the optimal value of zero. As another example, consider the discrete GAN shown in figure 4: here $M = 10,000$ and $N = 70,000$ so at any local minimum the generated images should be a linear combination of at least 7 training images. Indeed when we run OT-means with these values of $M, N$ we obtain images of low quality (shown in the bottom of the figure) even though the exact $W_1$ is better. This is reminiscent of the Mallasto et al. (2019) result in figure 1 but note that here we are using the exact $W_1$ between two discrete distributions and avoiding the intractable problem of approximating $W_1$ between continuous distributions.

Discrete FastGAN ($M = 10K$) ($W_1 \approx 1.82e04$):

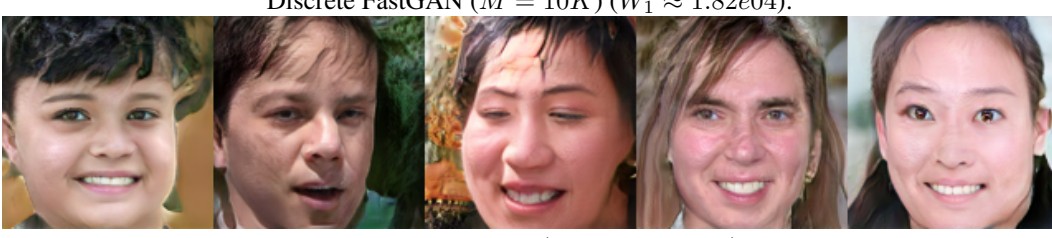

OT-Means ($M = 10K$) ($W_1 \approx 7.69e03$)):

Figure 4: Top: results of discrete fastGAN with $M = 10K$ so that $M < N$. The results are still sharp and realistic. Bottom: OT-means with the same $M, N$. This produces blurred images and yet the exact $W_1$ is lower.

## 3   THE IMPORTANCE OF THE DISCRIMINATOR ARCHITECTURE

Given the dual form of $W_1$ (equation 2), how do we explain the fact that WGANs do such a poor job of minimizing $W_1$? One possibility is that in the standard training of WGANs, the discriminator is not optimized to convergence: the common practice is to iterate a few iterations of gradient descent for the generator and then a few iterations for the discriminator. Perhaps this use of iterative gradient updates precludes the WGANs from optimizing $W_1$ ? In this section, we present evidence that this is not the case. In fact, even with iterative gradient updates, WGANs can do an excellent job of optimizing the Wasserstein distance, but the specific form of the distance that is being optimized is heavily influenced by the architecture of the discriminator.

Figures 5,6 show experiments with *fully connected discriminators*. Fully connected discriminators are interesting because they satisfy the universal approximation property Hornik et al. (1989): by using a sufficiently wide fully connected network with the gradient penalty method of WGANs, the discriminator should be able to implement any 1-Lipshitz function. In these figures, we considered three toy datasets of $N = 1000$ images: (1) white squares on a black background (2) MNIST and (3) face thumbnails (of size $64 \times 64$). We trained a discrete WGAN with a fully connected discriminator on these datasets with different values of $M$. The generator network was also a fully connected network and we used mini-batches of size 64. We also ran OT-Means on the same data and for the same values of $M$. We found that for a range of values of $M$, *the discrete WGAN did an excellent job of minimizing $W_1$*.

Figure 5 shows the results for $M = 64$. Recall that given theorem 2.4, any local minimum of the loss should generate images that are linear combinations of at least 15 different training images. Indeed when we look at the results of OT-Means (in the middle column) we find that the generated images are blurred, as expected. When we look at the results of the DiscreteWGAN-FC (right column) we see that they are also blurred. Unlike the typical published WGAN generated images, where the results are sharp and contain high-resolution detail (e.g. figure 1), now the results are visually similar to the results of OT-Means. Perhaps most convincingly, when we measure the exact $W_1$ between the generated images and the training set, we see that the DiscreteWGAN-FC samples achieve nearly the same $W_1$ as the OT-Means result (and significantly better than the $W_1$ obtained by randomly choosing a batch of $M$ examples from the data as the generated images).

Figure 6 shows a similar pattern of results for the case $M = N$. According to theorem 2.4, in this setting the optimal $W_1$ is obtained when the generator simply copies the training examples. Indeed such a solution is found rapidly using OT-Means. But more surprisingly, this solution is also found using the Discrete GAN: even though the generator never has direct access to the training set (only through a noisy gradient signal given by the discriminator) and is trained with mini-batches, it

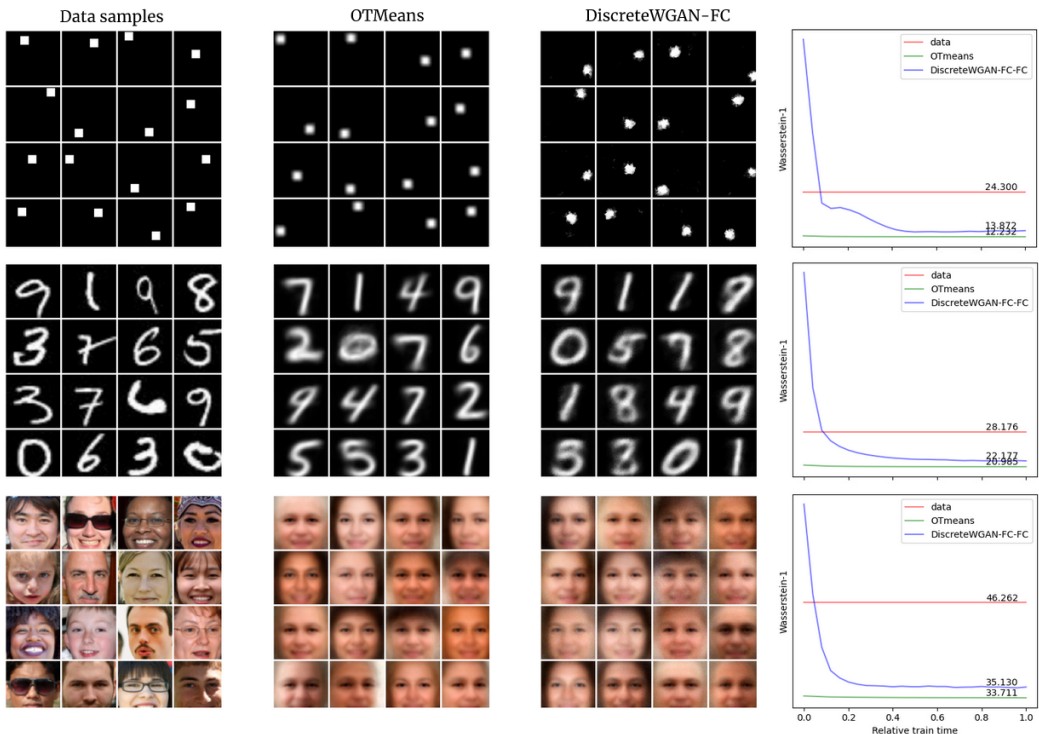

Figure 5: Using a fully connected discriminator minimizes W1 almost as well as OT-Means. The plots on the right show the exact $W_1$ for OT-Means (green), WGAN (blue), and randomly choosing a batch of $M$ training images (red).

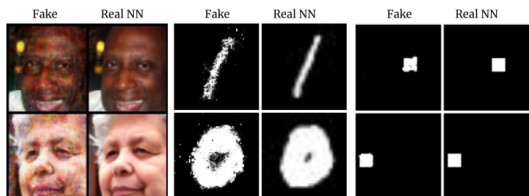

Figure 6: Using Fully connected discriminator and $M = N$ makes WGANs copy the data. Again, this is almost as good as minimizing W1 with OT-Means.

manages to copy the training examples and achieves a $W_1$ value that is close to the optimal value of zero. Figure 20 in the appendix shows more results for the case $M = N$: all images generated by the WGAN are copies of a training image.

Taken together, these results show that WGANs *can* do an excellent job of minimizing $W_1$, even with iterative gradient updates. How then do we explain the failure of $W_1$ minimization in successful WGANs such as those shown in figures 1,2? As we now show, this is because they use *convolutional discriminators*.

## 3.1 Convolutional Discriminators

The key assumption in the connection between WGANs and $W_1$ is that the discriminator can approximate any 1-Lipshitz function. But what happens if the discriminator architecture has a strong inductive bias? Almost all discriminators used in practical GANs are convolutional. The following theorems show that for such discriminators, WGANs no longer minimize $W_1$ between images but rather between smaller image-patches.

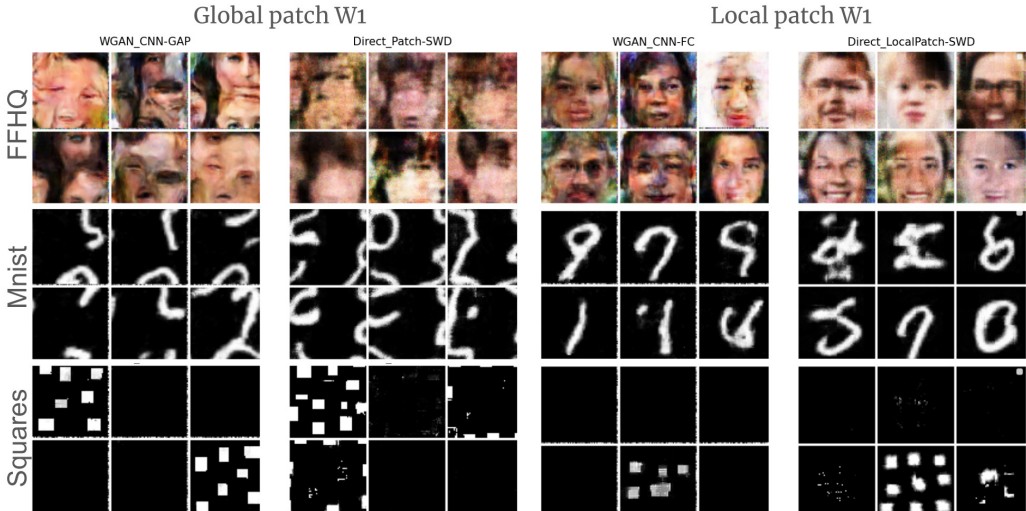

Figure 7: Using convolutional discriminators with the same generator as above learns either global patch statistics (with CNN-GAP) or local patch statistics (with CNN-FC). In all cases, the patch W1 that it finds is similar to directly optimizing the appropriate patch W1.

We start by formally defining two convolutional architectures (see detailed description in the appendix B.1).

**Definition 3.1.** (**CNN-GAP**) We denote by CNN-GAP(S) a CNN in which all layers except the last two are convolutional, followed by global average pooling (GAP) and a fully connected layer. The receptive field of units in the layer before the global average pool is $S$.

**Definition 3.2.** (**CNN-FC**) We denote by CNN-FC(S) a CNN in which all layers except the last layer are convolutional followed by fully connected layer and the receptive field of units in the layer before last is $S$.

These two CNNs are abstractions of CNNs that are used in successful WGANs: CNN-GAP is an abstraction of the patchGAN discriminator (Isola et al., 2017) and CNN-FC is an abstraction of the DCGAN discriminator (Radford et al., 2015). More details are in Appendix B.1.

**Theorem 3.3.** *Training a WGAN with a CNN-GAP discriminator is equivalent to minimizing $W_1(\hat{P}_\theta, \hat{P}_{data})$ where $\hat{P}_\theta, \hat{P}_{data}$ are the distribution over all* patches *of size S in the generated images and training images respectively.*

*Proof.* (sketch) For this discriminator the output can be written as a sum of discriminators over patches and by linearity of expectation $E_P(f), E_Q(f)$ can be written as a sum of expectations over patch distributions. Constraining the image discriminator to be 1-Lipshitz also constrains the patch discriminator. Full proof in A.2 □

This proof generalizes the connection between convolutional discriminators and patch distributions that was first presented in Isola et al. (2017). We note also that most of the proof (which relies on the linearity of expectations) also holds for other forms of GANs such as Cramer GANs (Bellemare et al., 2017) , Sobolev GANs (Mroueh et al., 2017), and MMD-GANs (Li et al., 2017) . For the non-saturating GAN with a convolutional discriminator, the discriminator output cannot be written as a linear sum of patch outputs and so the proof does not hold although we observe empirically that such NS-GANS behave qualitatively similar to WGANs (figure 17). Figure 19 shows the influence of the receptive field of the discriminator on the patch distribution minimized by the WGAN: as we increase the receptive field of the discriminator, the images increasingly capture more of the global structure of the training images.

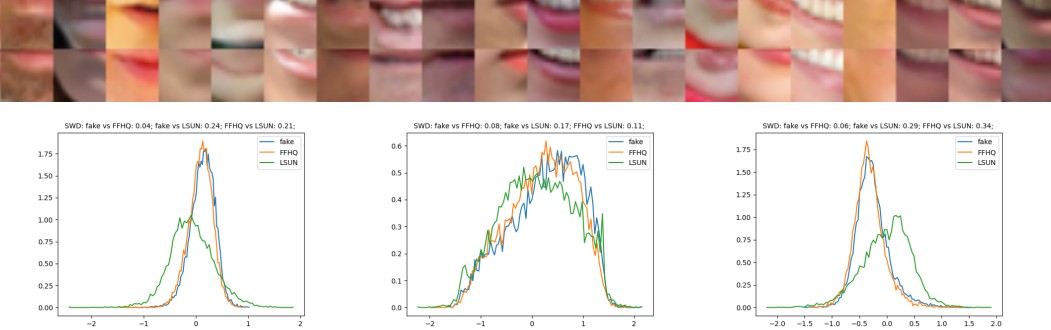

Figure 8: Top column: 20 randomly selected fake patches from a single location from a discrete GAN with $M = N = 70K$. Second column: the most similar true patches from the same location in the training set. Even though the generator never has direct access to the training images, it does a good job of copying the training patches. The mean cosine distance to the nearest neighbor is 0.99. Bottom: comparisons of histograms of random projections of fake and true patches at the same location. The histograms are almost perfectly aligned. The histogram of LSUNS patches at the same location are shown for comparison.

**Theorem 3.4.** *Training a WGAN with a CNN-FC discriminator is equivalent to minimizing an upper bound on $W_1(P_\theta^{\hat{i},j}, P_{data}^{i,j})$ where $P_\theta^{\hat{i},j} P_{data}^{i,j}$ are the distributions over all* patches *of size $S$ in the generated images and training images respectively at location $(i, j)$. This bound holds for any location $(i, j)$.*

*Proof.* (sketch) For this discriminator, the output can be written as a weighted sum of discriminators over patches at a specific location. By choosing the weights to be one only at one location and zero everywhere else, we obtain a bound on the Wasserstein distance between fake and generated patches at that location. Full proof in A.3. □

Figure 7 (1st and 3rd left columns) shows the dramatic influence of the discriminator architecture on the toy datasets that we showed in figure 5, Even though we are using exactly the same value of $M$ (64) and exactly the same training set and generator architecture, the Discrete GANs with CNN discriminators generate completely different images compared to OT-Means (Fig. 5).

In order to directly optimize patch $W_1$ we can no longer use OT-Means because the generated patches must satisfy the constraint that they are taken from $M$ images *with overlap*. As an alternative, we use SGD to train the same generator network used by the discrete GANs but replace the WGAN training loss with an estimate of the appropriate patch $W_1$.

As mentioned in the introduction we use the discrete generators to avoid using minibatch-based empirical estimates of $W_1$. For image $W_1$ we need to compute an optimal transport between $M = 64$ generated images and $N$ training images and this is still feasible but for global patch $W_1$ we need to compute optimal transport between all patches in 64 images and all patches in the training images and this is infeasible. We therefore used Sliced Wasserstein distance (SWD) Pitie et al. (2005); Rabin et al. (2011); Bonneel et al. (2015) as a cheaper proxy.

The sliced Wasserstein distance is defined as the expected $W_1$ value between 1-d projections of two distributions into random directions. An unbiased approximation of this distance is computed from k random directions. The efficiency of SWD stems from the fact that for one-dimensional data, the $W_1$ can be computed by simply sorting the samples and so the distance between two samples of size $M$ can be computed in $O(M \log M)$. While in general $SWD(P, Q)$ and $W_1(P, Q)$ may be different, SWD shares with $W_1$ the property that $SWD(P, Q) = 0$ if and only if $P = Q$. Thus by minimizing patch $SWD$ we are encouraging the patch distribution in the fake images to match the patch distributions in the true images.

Figure 7 (2nd and 4th columns from the left) shows the results of direct optimization of patch $SWD$ on the same datasets and using the values of $M$. Unlike the OT-Means results, matching patch

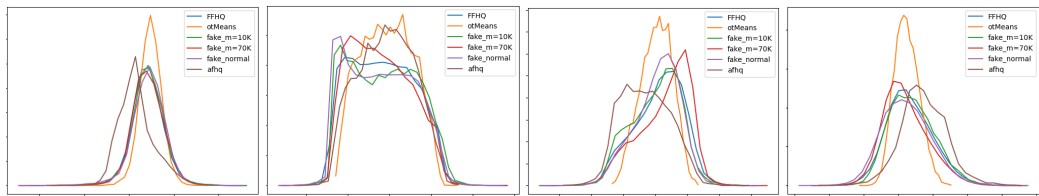

Figure 9: Evidence that the discrete DCGAN (2 leftmost plots) and FastGAN (2 rightmost plots) match the local statistics in the generated image and the training images. Even though each image is quite far from a training image, the local histograms of random projections of patches from the true images and the generated images are almost identical. This is true for various values of $M$ including a continuous prior.

distributions yields images that preserve high-frequency details and lack global structure. Clearly, discrete GANs that use convolutional discriminators generate images similar to those obtained by approximately minimizing patch distributions, not by minimizing image $W_1$.

To summarize, our proofs show that theoretically WGANs with convolutional discriminators should minimize patch $W_1$ and not image $W_1$ and our experiments with toy data show that this is indeed the case, even if we use iterative gradient training. We now ask, can the minimization of patch $W_1$ explain the success of WGANs in realistic datasets?

# 4  DO SUCCESSFUL DISCRETE GANS MINIMIZE PATCH WASSERSTEIN DISTANCE?

| Model | DCGAN | | | FastGAN | | | OT-Means | AFHQ | Imagenet |
|---|---|---|---|---|---|---|---|---|---|
| **M** | 10K | 70K | $\infty$ | 10K | 70K | $\infty$ | 10K | 15K | 70K |
| **Local patch swd** | 0.044 | 0.040 | 0.058 | 0.056 | 0.045 | 0.033 | 0.099 | 0.145 | 0.181 |

Table 1: The local SWD between real FFHQ (N=70K) image patches and fake image patches for different successful DiscreteWGANs (averaged over locations) with different M values. In all cases, the average local SWD is below 0.05. For reference comparison, the average local SWD for OT-Means (M=10K) is around 0.1 and $> 0.14$ for images from AFHQ and ImageNet

We trained discrete versions of FastGAN and DCGAN on $N = 70,000$ FFHQ images of size $128 \times 128$. We varied the number of fixed noise vectors $M$ and found that for both architectures, the discrete GANs generate comparable results to the continuous counterparts (figures 2,4). See also figure 14 in the appendix.

The preceding theory and experiments have told us what to expect if these successful discrete GANs are minimizing Wasserstein distance. The easiest case is when $M = N$. For such a case, we expect a discrete GAN to copy images from the training set if it is minimizing image $W_1$ and to copy patches from the training set if it is minimizing patch $W_1$. Figure 8 (top) shows fake patches from a single location generated by the discrete FastGAN from figure 2 and the bottom row shows the closest match from the same location in the training set. Even though the generator never has direct access to the training images, it does a good job of copying the training patches. The mean cosine distance to the nearest neighbor is $0.99$.

If the discrete GAN is minimizing patch $W_1$ it is not enough for it to copy patches from the training set: it should also maintain the same patch distribution. Consider for example a location where there are only two possible patches in the training set: $60\%$ are blue patches and $40\%$ are green patches. If the fake patches are $99\%$ blue and $1\%$ green, then the patch $W_1$ would still be high (Elnekave & Weiss, 2022). Only if the fake patches were also $60\%$ blue and $40\%$ green is the patch $W_1$ minimal. To visualize how well the discrete GAN matches the local patch distribution we considered histograms of random projections of the patches. As can be seen in three plots at the bottom of figure 8 the histograms for fake patches and true patches are almost perfectly aligned. Thus the discrete GAN is not only copying patches but also matching the local patch distribution. Even when we consider the histograms corresponding to patches in the edges of the image for true and fake

images, the two histograms align quite well (see figure 13 in the appendix). Thus even for locations that often correspond to the background, the discrete GAN matches the local patch distribution in the true and fake images.

When $M < N$ then we cannot expect the discrete GANs to copy training patches but rather its output should look like the output of OT-Means on patches from a single location (see appendix D): each generated patch is a linear combination of training patches and the local histograms of true and fake patches at a location should match. Figure 9 shows that the local distribution is matched very well for a discrete DCGAN with different values of $M$. Importantly, the matching also holds with the continuous prior.

We summarize all of our experiments in table 1: for the two GAN architectures and for different values of $M$, the local distributions at a particular location are well matched between the true images and the generated images. As a reference we also compute the distance to a batch of 10K images from ImageNet and a batch from AFHQ (Choi et al., 2020): the average local SWD between AFHQ and FFHQ is about $0.15$ and $0.18$ while the different WGANs consistently achieve a local SWD that is about $0.04$.

## 5 LIMITATIONS AND EXTENSIONS

Our theoretical results are based on simplifications of convolutional discriminators used in practice. One major simplification is that we assume that the receptive field sizes are at most $S$ and this value of $S$ determines the patch size used in patch $W_1$. For practical CNNs, the theoretical limit on the receptive field may be very large, but the effective receptive field size is still small (Brendel & Bethge, 2019) and the CNN can be well-approximated as if the receptive field size was much smaller than the theoretical limit. A second simplification is that we implicitly assumed that the CNNs do not use any padding when performing convolutions (and this was also the case in the GANs that we trained for the toy data). When padding is used, patches at different image locations can actually be distinguished even with a patch-based discriminator (e.g. (Isola et al., 2017),(Shaham et al., 2019)) so even a CNN-GAP discriminator will optimize a patch distance that is location-dependent.

Even though our use of the discrete setting allows us to exactly measure $W_1$ between true and fake images, it still leaves open the question of how small $W_1$ needs to be in order for a method to be described as "successful" in minimizing $W_1$. For this reason, we compare the results of the WGANs to direct optimization of $W_1$ and also visualize the histograms and measure their distance using SWD.

The most significant limitation of our paper, of course, is our focus on discrete GANs while SOTA GANs use a continuous prior. We note that most of our theoretical results do not require the GANs to be discrete: working with discrete GANs allows us to empirically measure $W_1$ and rigorously determine whether the predictions hold. In practice, we find that continuous GANs with convolutional discriminators behave similarly to their discrete counterparts (with large $M$).

## 6 DISCUSSION

The question that motivated this paper is whether WGANs succeed because they optimize the Wasserstein distance. We leveraged the discrete setting that allows us to compute the Wasserstein distance exactly, to characterize its optima, and to optimize it with alternative algorithms. Our results indicate that the answer is "yes" but the form of $W_1$ that is being optimized depends on the architecture of the discriminator. Specifically, we have shown that when the discriminator is convolutional, what is being minimized is the patch $W_1$ and not the image $W_1$.

A major advantage of having a well-defined loss function that is being optimized is the ability to monitor learning algorithms and check for their convergence. We hope that our results will yield WGAN learning algorithms that are considerably more stable than the current methods and require less parameter tuning.

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

## A  PROOFS

### A.1  PROOF OF THEOREM 2.4

To show that for $M < N$ at any local minimum, the generator must be a linear combination of at least $N/M$ training examples, we differentiate equation 3 with respect to a generated image $x_i$ and setting the gradient to zero yields that $x_i$ must satisfy:

$$x_i = \sum_j w_{ij} y_j$$

with $w_{ij}$ a vector of weights that sums to one and $w_{ij} \propto \pi_{ij} \|x_j - y_i\|^{-1}$. Now since $\pi_{ij}$ is a solution to the optimal transport between $M$ generated images and $N$ training images, it is a non-negative matrix that satisfies $\sum_i \pi_{ij} = \frac{1}{N}$ and $\sum_j \pi_{ij} = \frac{1}{M}$. This implies that for each $i$ there must be at least $N/M$ different indices $j$ for which $\pi_{ij} > 0$ and this shows that $y$ must be a linear combination of at least $N/M$ training images.

### A.2  PROOF OF THEOREM 3.3

We can write the output of the critic for an input x as:

$$f(x) = W^T \frac{1}{n} \sum_i h(p_i x)$$

where $p_i x$ extracts the $i$th patch in image $x$, $W$ are the weights in the final layer, and $n$ is the number of patches in the penultimate feature map.

Moving $W^T$ into the above sum and defining $g(x) = W^T h(x)$ where $x$ is a patch we can rewrite the image critic, $f(x)$ as:

$$f(x) = \frac{1}{n} \sum_i g(P_i x)$$

Recall that the critic $f$ attempts to maximize $E_P(f) - E_Q(f)$. By the linearity of the expectation $E_Q(f)$ is equal to $E_{\tilde{Q}}(g)$ and likewise $E_{P_\theta}(f) = E_{\tilde{P}_\theta}(g)$ where $\tilde{P}, \tilde{Q}$ are the distributions over patches in the true and fake images. Denoting by $GAP_1$ the class of 1-Lipshitz functions that can be implemented by a CNN-GAP architecture, this means that:

$$\max_{f \in GAP_1} E_{P_\theta}(f) - E_Q(f) = \max_{g \in G_1} E_{\tilde{P}_\theta}(g) - E_{\tilde{Q}}(g)$$

$G_1$ is the class of functions that operate on $S \times S$ patches and can be implemented by the units in the layer before the GAP.

Constraining the image discriminator to be 1-Lipshitz means that for any two images $x_1, x_2$ such that $\|x_1 - x_2\| < d$, $|f(x_1) - f(x_2)| < d$. In particular, this holds when we choose $x_1, x_2$ so that they differ only in a single patch and this means that $g$ must also be 1-Lipshitz.

### A.3  PROOF OF THEOREM 3.4

We can write the output of the critic for an input x as:

$$f(x) = \sum_c \sum_i w_{ic} f_c(P_i x) \tag{4}$$

where $P_i x$ extracts the $i$th patch in image $x$, $C$ is the number of channels and $w_{ic}$ the weights in the final layer.

Now define the function class $CNN$ as all functions that can be implemented using equation 4 and the subclass $CNN_k$ as the set of functions that can be implemented by equation 4 where $w_{jc} = 0$ $\forall c$, $\forall j \neq k$. Since $CNN_i \subset CNN$ we have

$$\max_{f \in CNN} E_{P_\theta}(f) - E_Q(f) \geq \max_{f \in CNN_i} E_{P_\theta}(f) - E_Q(f)$$

and since for $f \in CNN_i$ f is only a function of the $i$th patch we can write:

$$
\begin{aligned}
\max_{f \in CNN_i} E_{P_\theta}(f) - E_Q(f) &= \max_{i \in I, g_i \in G} E_{\tilde{P}_\theta}(g_i) - E_{\tilde{Q}}(g_i) \\
&= \max_{i \in I} W_1^{patches_i}(P_\theta, Q)
\end{aligned}
$$

where again $g_i(x)$ is a critic for the $i$th patch: $g_i(x) = \sum_c w_{ic} f_c(P_i x)$.

## B  ARCHITECTURES

### B.1  TOY CNNS

We bring here further details about CNN-GAP and CNN-FC architectures used in the experiment section. Both architectures start with 3 convolutional layers with kernel size 3, stride 2 and no padding each followed by a RELU layer. These layers transform a 64x64x3 image into a 7x7x256 feature map. The receptive field at this point, i.e., the size of the patch in the original images that affect each pixel in this feature map is 15x15. CNN-GAP applies global average pooling to this feature map that transforms it into a 1x1x256 layer that is later linearly projected into a scalar. CNN-FC reshapes the feature map into a long vector of 7*7*256 that is linearly projected into a scalar. Figure 10 illustrates the two architectures with their common backbone.

### B.2  FASTGAN VARIANT

We experimented with FastGAN Liu et al. (2020) as an easy, fast-to-converge GAN model that performs comparably to SOTA GANs. In our experiments, we simplified its architecture branching off this implementation [2]. We removed the auto-encoding branches of the discriminator so that both the generator and the discriminator are simple feed-forward networks with skip connections.

## C  OTMEANS WITH DIFFERENT VALUES OF $M$

In the main paper we showed results for small $M$ (in which case the generated images are blurred) and $M = N$ (in which case the generated images are copies of the training set). In figure 11 we show intermediate values of $M$ as well. As can be seen, as $M$ increases, the images are increasingly sharper until they appear to copy the training set.

## D  OTMEANS ON PATCHES

One reason to expect better results with the minimization of patch $W_1$ is related to theorem 2.4: minimizing image $W_1$ causes the generator to either copy the training set or to generate images that are linear combinations of several training images and will therefore be of poor visual quality. But if we minimize patch $W_1$ we can still generate images that are very different from the training images by copying patches from the training set and combining them in novel ways. We can also generate patches that are linear combinations of training patches, but since the distribution of patches is simpler in some sense than that of images, linear combinations of training patches do not necessarily

---

[2]https://github.com/odegeasslbc/FastGAN-pytorch/blob/main/models.py

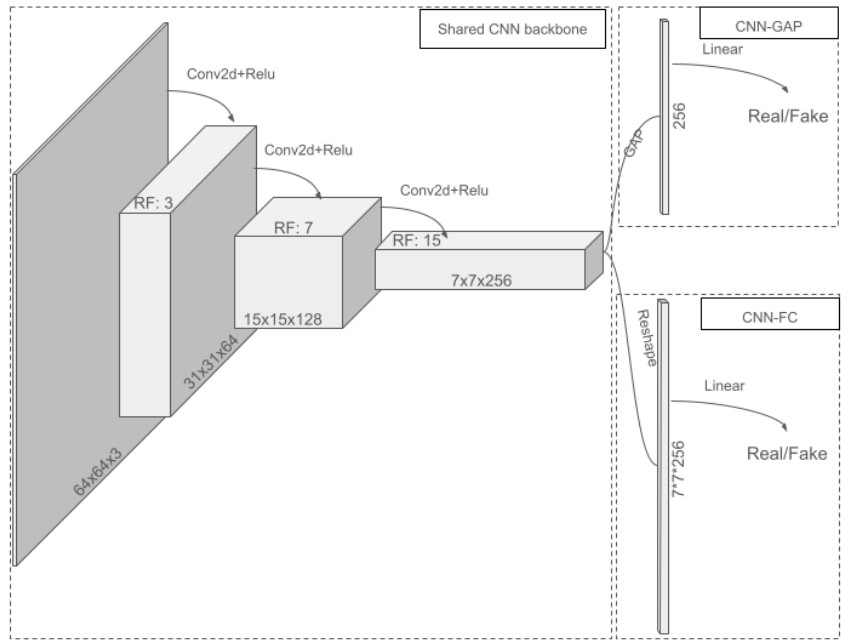

Figure 10: An illustration of the toy CNN used the experiments in figures 5-7. 3 Convolutional layers are followed by two types of linear heads that project the feature map into a scalar. The annotation **RF** stands for the receptive field of the previous layer.

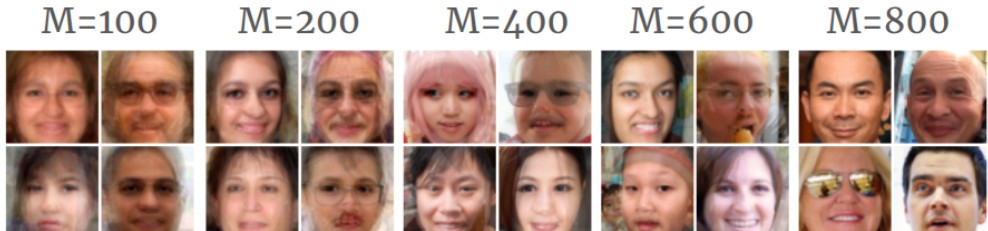

Figure 11: Results of OTmeans with $N = 1000$ and different values of $M$. As $M$ increases, the images are increasingly sharper until they appear to copy the training set.

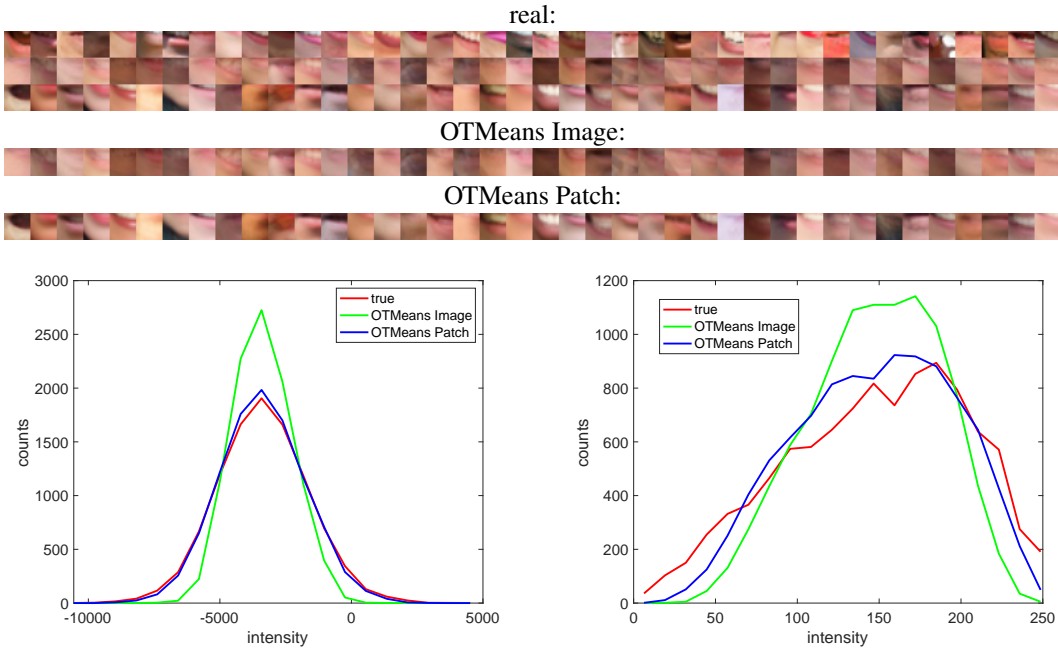

Figure 12: The difference between minimizing image $W_1$ and patch $W_1$. When we minimize image $W_1$ and then extract small ($20 \times 20$) patches (middle row) the extracted patches look very different from the training patches. When we minimize patch $W_1$, i.e. the Wasserstein distance between $20 \times 20$ patches at that location in the true and fake images, the generated patches (bottom row) look similar to real patches. The success in capturing the local patch distribution can be visualized by comparing the histograms of random projections of the patches.

have to be blurred. Figure 12 illustrates this point. We minimized patch $W_1$ by running OTmeans with $M = 10,000$ only on a small 20x20 patch in a location that corresponds to the left part of the mouth. We then compared the generated patches to those obtained by running OTmeans on the full image (hence minimizing image $W_1$) and then extracting the patches. As can be seen in the figure, the patches obtained by minimizing patch $W_1$ are high-contrast and sharp, unlike the patches obtained by minimizing image $W_1$.

We can visualize how well the generated patches match the distribution of training patches by plotting the histogram of random projections of the patches. As shown in the bottom of figure 12, when we minimize image $W_1$ the histograms of projections of generated patches are visibly different from those of training patches at the same location. When we minimize the local patch $W_1$, the histograms match very well.

## E    HISTOGRAMS AT DIFFERENT LOCATIONS

Figure 13 visualize how well the discrete GAN matches the local patch distribution we considered histograms of random projections of the patches in additional 3 locations.

## F    RESULTS WITH DISCRETE SOTA ARCHITECTURES

We bring here the results of training discrete-WGANs with more capable architectures. As can be seen, for both DCGAN Radford et al. (2015) and FastGAN Liu et al. (2020) the results with two different discrete priors (M=10K,70K) are comparable to the results with normal prior (M=∞). Each second row shows the data nearest neighbor to show there is no copying involved.

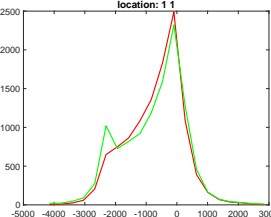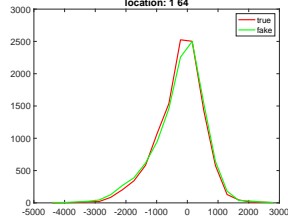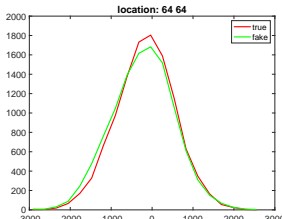

Figure 13: Histograms of the same random projection for true and fake patches at different locations. Note that the histograms are different between different locations but highly similar for true and fake patches. This is true even for locations at the edge of the image.

### F.1   RESULTS WITH DISCRETE DCGAN

See generated images and data nearest neighbors in figure 14

### F.2   RESULTS WITH DISCRETE FASTGAN

See generated images and data nearest neighbors in figure 15

## G   ADDITIONAL ABLATION EXPERIMENTS

### G.1   RESULTS WITH CONVOLUTIONAL GENERATOR

In all of the experiments in the paper, we used the same FC generator. We repeat here the results from figures 5 and 8 from the paper with a convolutional generator (DCGANRadford et al. (2015). As can be seen in figure 16 the WGAN and direct optimization show similar behavior.

### G.2   RESULTS WITH DIFFERENT GAN LOSSES

Our paper deals with WGANs. While the same can be done for all IPMs like Sobolev GANs and MMD GANs it may not be directly applied to some other losses like the original or non-saturating GAN losses. However, our experiments with Non-saturating GANs show (figure 17) qualitatively similar results.

### G.3   RESULTS WITH DIFFERENT PATCH SIZES

We have conducted experiments from figures 5,7 with different receptive field sizes for the discriminator. We used 2 convolutional layers instead of 3 in the discriminator to get RF size=8 and the same number of layers (3) with kernel size 4 (instead of 3) to get a RF size of 22. As can be seen in figure 19, as the receptive field increases the WGAN generated images preserve statistics of patches of larger sizes.

### G.4   ADDITIONAL RESULTS WITH M=N

We bring here in figure 20 more results from the experiment of figure 6

$M = 10,000:$

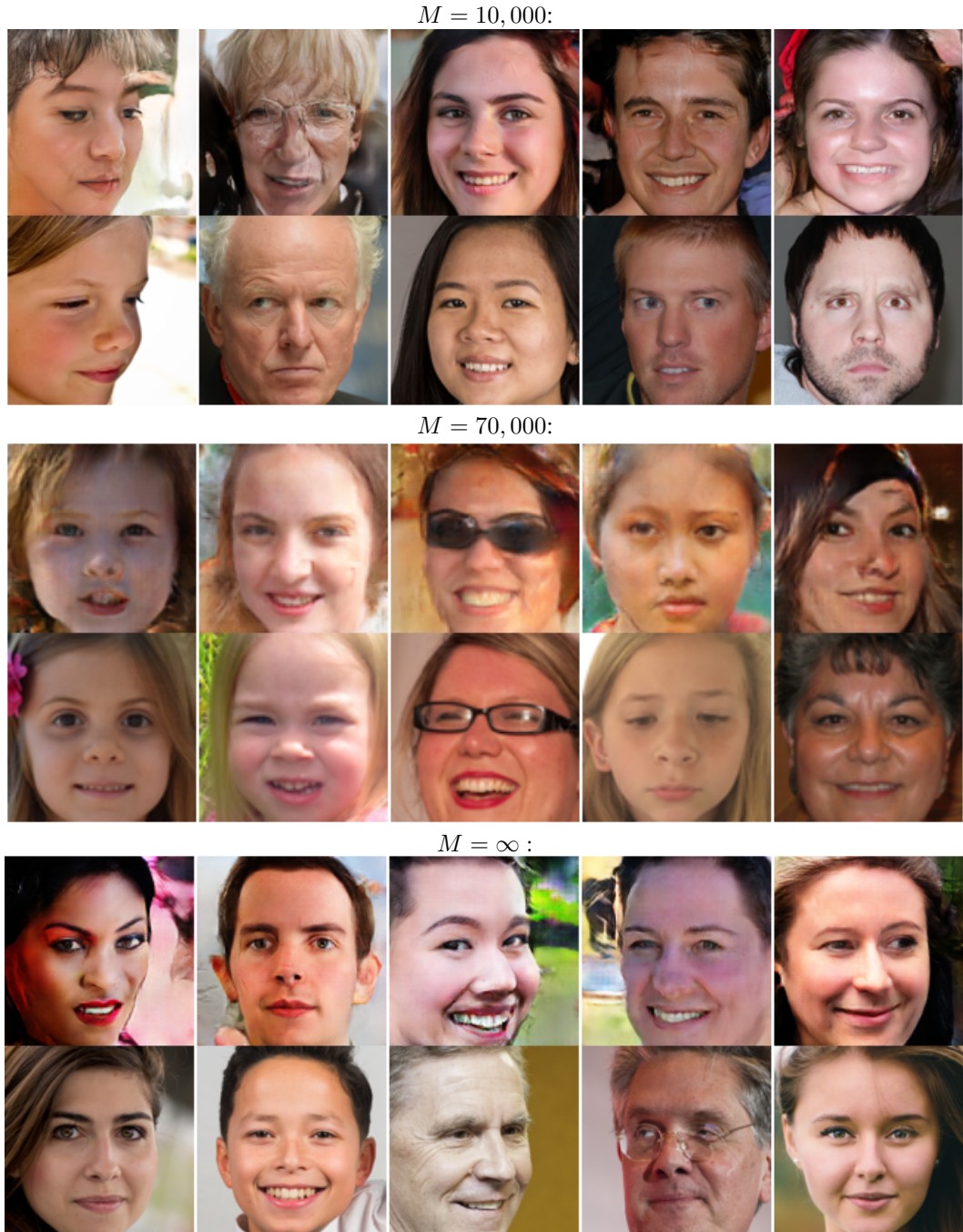

$M = 70,000:$

$M = \infty:$

Figure 14: Images generated with Discrete DCGAN for different values of $M$. For each value of $M$ we show five generated images and below them the closest image from the training set for each generated image.

$M = 10,000$:

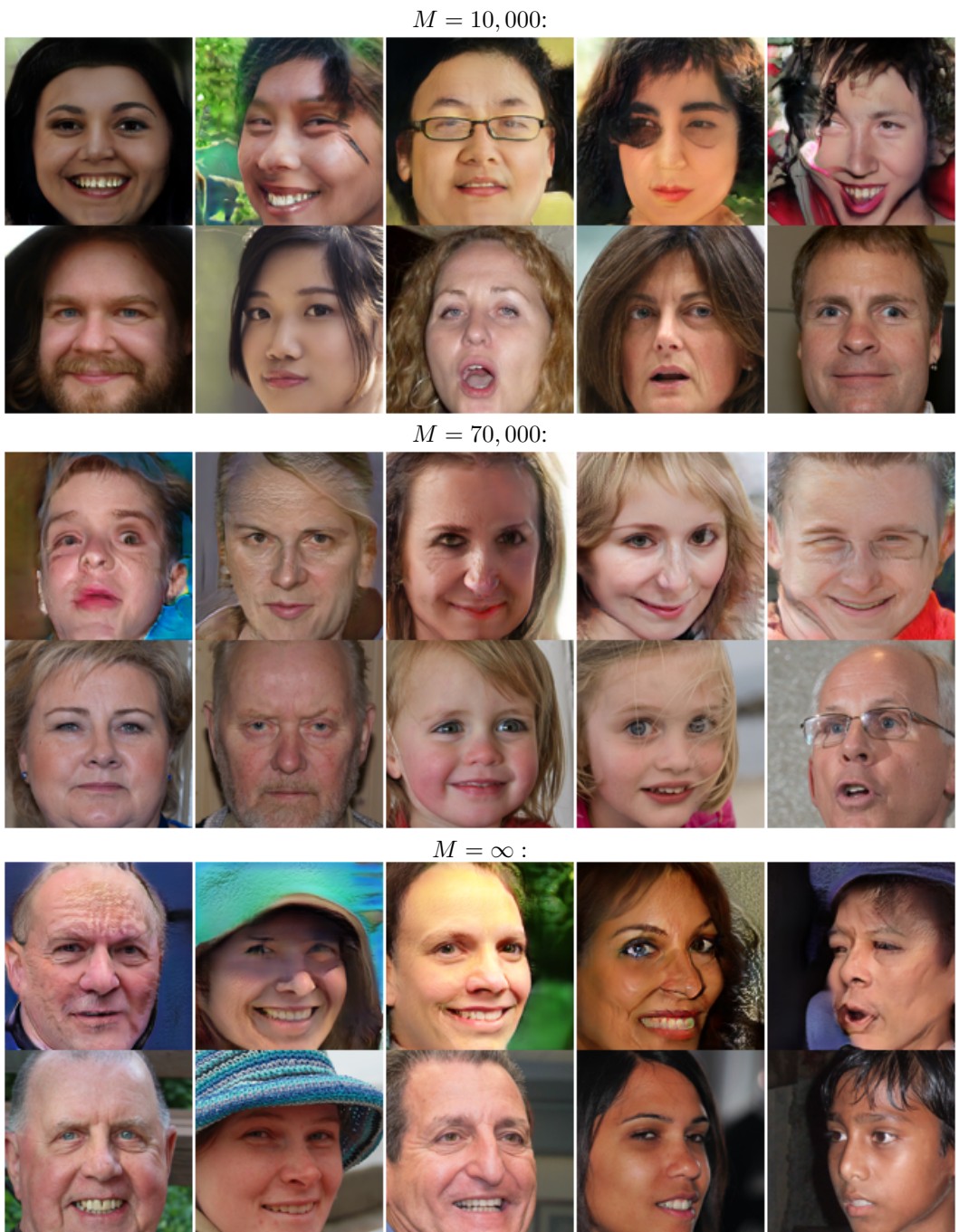

$M = 70,000$:

$M = \infty$:

Figure 15: Images generated with Discrete FastGAN variant for different values of $M$. For each value of $M$ we show five generated images and below them the closest image from the training set for each generated image.

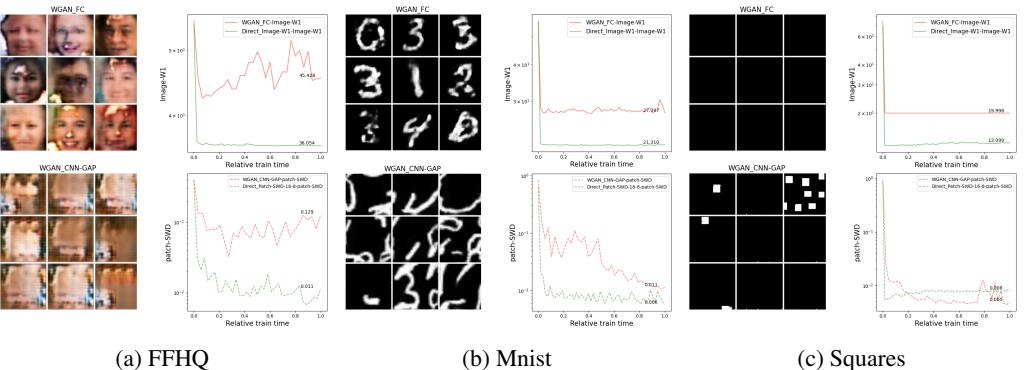

Figure 16: Comparing direct optimization in image/patch level with WGAN with FC/Convolutional discriminator when the generator is convolutional

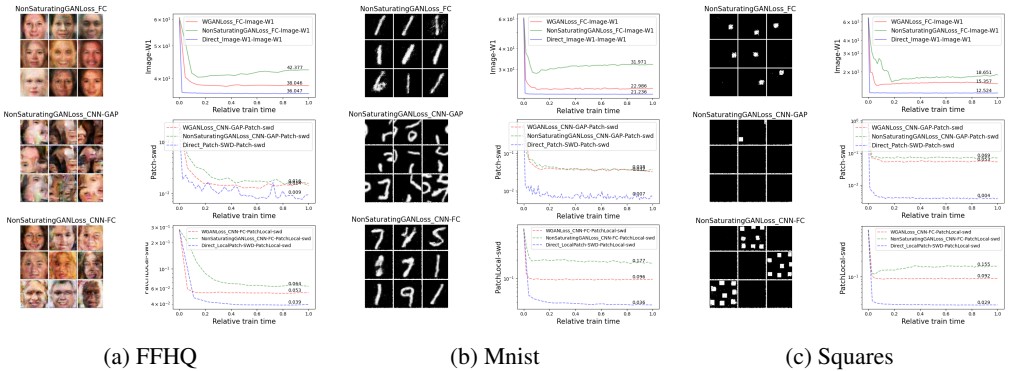

Figure 17: Results of training NS-GAN with FC/Convolutional discriminators. As can be seen, we see the same trend as with WGAN where the discriminator architecture controls whether the statistic being preserved is in the image or the patch level. The graphs also show metrics taken from the same model trained with WGAN loss and of a direct optimization for reference. The NS-GAN is not as good as the WGAN at minimizing the appropriate $W_1$ but the results are qualitatively similar.

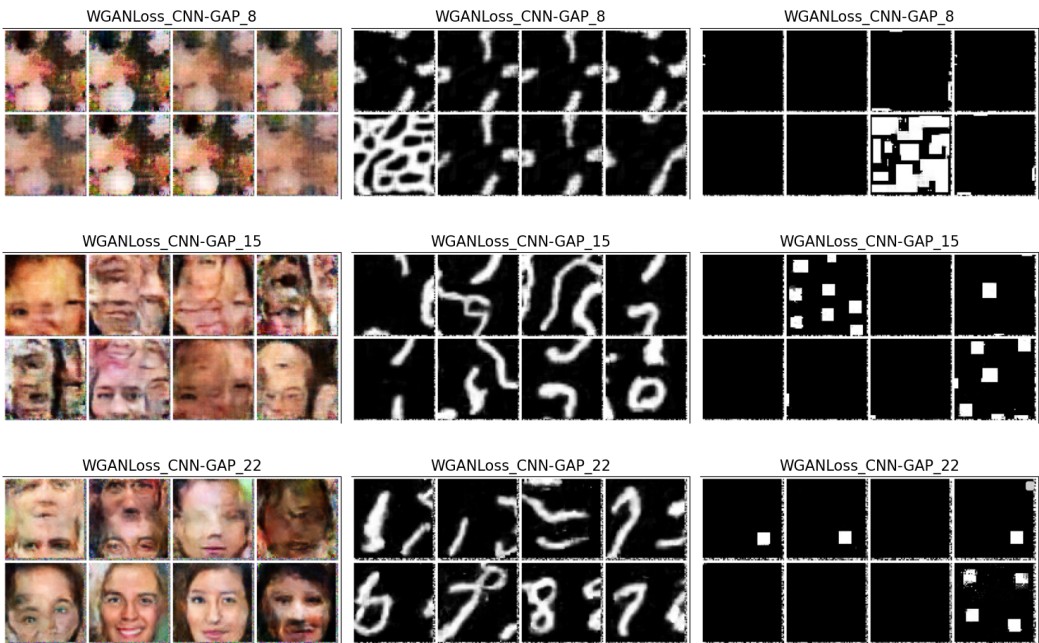

Figure 18: CNN-GAP with different receptive fields

Figure 19: Results of DiscreteWGAN trained with Convolutional discriminator with GAP but with different receptive fields. As can be seen, as the patch size grows the statistics of bigger patches are preserved.

*Batch*1:

*Batch*2:

*Batch*3:

*Batch*4:

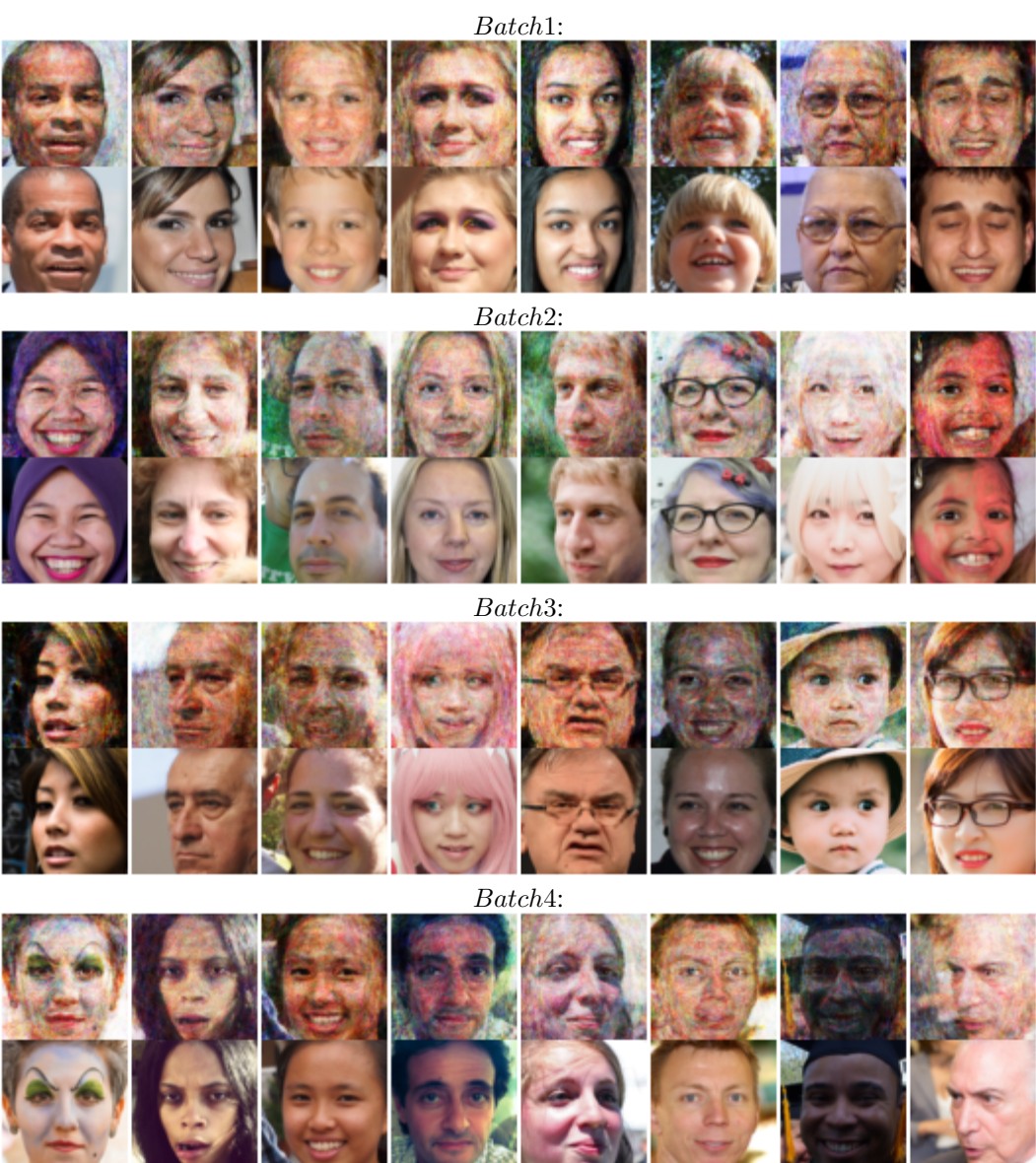

Figure 20: Additional 4 batches of results from the model from figure 6 trained with M=N=1K on FFHQ each batch's top row shows fake images and the bottom row shows data nearest neighbors. All generated images are copies of training images.

