# OpenReview forum: "Do WGANs succeed because they minimize the Wasserstein Distance? Lessons from Discrete Generators"
_ICLR.cc/2025/Conference — ICLR 2025 Poster_

### Official Review · Reviewer_1sSp · 2024-10-26

**Soundness:** 3
**Presentation:** 3
**Contribution:** 3
**Rating:** 6
**Confidence:** 4

**Summary:**

In this paper, the authors propose a framework to analyzing Wasserstein GANs, and in particular, understanding if the model truly minimizes the W1 loss while training. The insights derived help explain the visual quality of the images generated by certain GAN architectures, and also pave the way for similar analysis of other GAN variants.

**Strengths:**

- The presentation of the paper is good and the writing is easy to follow
- The overall results derived make sense, and correlate well with the observed performance of existing patchGAN and DCGAN-style architectures.
- The theoretical formulation is sound, to the best of my reading. The results also make intuitive sense.
- I am also able to corroborate when I read, with the empirical evidence I’ve seen while training GANs, which, in my view, suggests that the paper succeeds at what it does

**Weaknesses:**

- While the analysis carried out is good, I do find it to be somewhat lacking in breadth. It might be good to see how this theory could be applied outside of the Lipschitz-constraint-based W1 loss. For example, could one draw insights into the various gradient regularization strategies that people have used to approximate W1 (i.e., the Sobolev spaces). E.g., [1,2,3] (just to name a few), where we observe very similar artifacting to the ones derived and analyzed in this paper. Would analyzing such other discriminator architectures/losses yield a more holistic view of the space of WGANs?

- The ablation concerning the Convolutional GANs seemed lacking. Given such a strong correlation between the receptive field of the convolution layers and the patch-based W1 minimization, some insights into how we control or interpret this value would be useful for future design. While I understand from Section 5 that estimating S is not practical for DCGAN-style architectures, maybe toy experiments involving a single convolution layer might show links between the filter size of the convolution and S.

- There have many other works that propose and analyze well-defined loss function that allow one to monitor learning algorithms and check for their convergence (See [4,5,6]). It might be worth discussing such approaches too in the related works.

- Overall, I still feel that the insights developed by this paper outweigh the weaknesses mentioned, and some of them, such as generalizing to other losses could be discussed now, but more thoroughly explored in future works. I am therefore inclined towards an accept.

======

[1] “Banach WGANs,” Adler and Lunz,

[2] “Demystifying MMD GANs,” Binkowski et al.

[3] “Coulomb GANs,” Unterthiner et al.

[4] “Euler-Lagrange Analysis of GANs,” Asokan and Seelamantula

[5] “Sobolev GANs,” Mroueh et al.

[6] “How Well Generative Adversarial Networks Learn Distributions,” Liang

**Questions:**

See Weaknesses

---

> ### Author Response · Authors · 2024-11-19
> **Answer to Reviewer 1sSp**
>
> Thank you for your constructive comments.
>
> We agree that the analysis can be extended to additional GANs. As we noted in the response to reviewer beHd, the main step in the proof is to use the linearity of expectation to convert  E_p(f)-E_q(f) into a sum of expectations over local patches. Thus for a large family of IPFs this part of the proof can be directly applied. Where we need to be a bit more careful is in showing that the regularization of the image critic leads immediately to a regularization of the patch critic, and this needs to be shown on a case-by-case basis. We will add this discussion to the final version.
>
>
> We have conducted experiments where we use less convolutional layers in the discriminator and observed that the relevant patch size indeed changes as predicted. [This figure](https://postimg.cc/mtzVbgFv) (and appendix F.3 in the updated version) compares WGANs trained with CNN-GAP discriminators of different depth and thus different receptive fields. As can be seen, with a shallow discriminator the generated images preserve statistics of smaller patches. We will discuss these results in the final version.
>
> Thank you for your suggestion. We will add these references to the final version and discuss them.

---

### Official Review · Reviewer_bEHd · 2024-10-27

**Soundness:** 4
**Presentation:** 4
**Contribution:** 3
**Rating:** 6
**Confidence:** 5

**Summary:**

The Wasserstein GAN (WGAN) is a generative model that attempts to minimize the Wasserstein distance between the distribution of generated samples and the distribution of the training data. Several papers have pointed out that successful WGANs don't seem to minimize the Wasserstein distance. These papers further claimed that this is actually a good thing, as minimizing the Wasserstein distance would lead to blurry generated images.
This paper presents more rigorous experiments that support the conclusion from previous works. This is done by training discrete WGANs - models that are constrained to generate one out of M images - which allows to precisely compute the Wasserstein distance between the distribution of the GAN outputs and the empirical distribution of the training set.
A key observation in the paper is that when the discriminator is a convolutional network, WGANs minimize the Wasserstein distance between the distribution of patches of generated images and the distribution of patches of the training images. This is while when the discriminator is fully connected, WGANs do minimize the Wasserstein distance over whole images.

**Strengths:**

- The systematic analysis using discrete distributions is clever and makes a lot of sense.

- The theoretical characterization of the solution minimizing the Wasserstein distance in the the discrete case is nice.

- The paper does a pretty good job of empirically demonstrating the main claim, which is that WGANs with convolutional discriminators don't minimize the Wasserstein distance between distributions of whole images, but rather between distributions of patches.

- The paper is well written. Arguments are easy to follow.

**Weaknesses:**

- The main takeaway message that the paper highlights is the fact that WGANs with convolutional discriminators minimize the Wasserstein distance between patch distributions. The paper doesn't provide sufficient context and discussion about why this observation is novel. As stated in the paper, Isola et al. (2017) called GANs with a convolutional discriminator "patch GANs", suggesting they minimize distances between patch distributions. This was also explicitly stated by Rott-Shaham et al. (2019). In fact, the origins of these GANs can be traced back to [1] which called them "Markovian GANs", and to [2,3] which called them "spatial GANs". The latter two papers explicitly expressed the loss as a sum of GAN losses over patches of the size of the receptive field, which implies that these GANs attempt to minimize distances between patch distributions and not whole images. From that standpoint it could seem to the readers that this fact is well known and also that it is not unique to WGANs, but rather applies to any GAN. Nevertheless, I believe that this might not be a material weakness, but rather a weakness in the exposition in the paper (see Questions section below).

- The topic is not timely. The popularity of GANs has constantly decreased over the last few years, as diffusion models gained popularity. I consider this to be a minor weakness but I do believe that it affects the potential impact of the paper.

[1] C. Li and M. Wand, "Precomputed real-time texture synthesis with Markovian generative adversarial networks", ECCV`16.

[2] N. Jetchev, U. Bergmann, and R. Vollgraf "Texture synthesis with spatial generative adversarial networks", NIPS 2016 adversarial learning workshop.

[3] N. Jetchev, U. Bergmann, and R. Vollgraf, "Learning texture manifolds with the periodic spatial GAN", ICML`17.

**Questions:**

Regarding the first weakness stated above, I would be happy to hear the authors' thoughts about the following. As opposed to [2,3], which explicitly attempt to minimize a loss that is the sum of GAN losses over patches, standard GAN training applies the GAN loss after the pooling in the discriminator. In that case, the loss can be expressed as a sum of GAN losses over patches (i.e. swapped with the pooling operation) only if the loss is linear. This is the case for WGANs but not for other types of GANs. Therefore, other types of GANs with a convolutional discriminator don't directly minimize distances between patch distributions. This is while WGANs do. Is that correct?

If this statement is correct, then the paper would benefit a lot from emphasizing and discussing it. Otherwise, as stated above, it seems that the main claim in the paper is well known and not specific to WGANs (namely, it seems that all types of GANs with a convolutional discriminator minimize distances between patch distributions).

---

> ### Author Response · Authors · 2024-11-19
> **Answer to reviewer bEHd**
>
> Thank you for your constructive feedback. We appreciate your comments and suggestions, which will help improve our work.
>
> ## Novelty
>
> You are certainly correct that previous work has shown how to design convolutional discriminators that will explicitly minimize patch W1. Although we cite these papers, we agree that we should do a better job of describing their contribution in this regard.
>
> However, we think our result is significantly stronger. We show that WGANs that claim to be minimizing image W1 actually minimize patch W1 when a convolutional discriminator is used. Thus in the original WGAN paper and in the “Improved training of  WGAN” paper, all the results with images were obtained using the DCGAN discriminator and yet the theoretical part of the paper deals with minimizing image W1. Our theorem 3.4 shows that this type of discriminator (CNN-FC) is minimizing an upper bound on  the local patch W1. This should be contrasted with WGANs that use fully connected discriminators and actually do minimize image W1 (leading to blurred images or copies). In other words, just because you are using a discriminator that outputs a single number that represents the “fakeness” of a given image, doesn’t mean that you are minimizing image W1 and in fact you might be minimizing patch W1 depending on the particular architecture.
>
> Additionally, as you noted in your review, our paper also uses the discrete setting to show experimentally that WGANs indeed minimizes the appropriate W1. To the best of our knowledge, this has not been shown in previous works.
>
> ## Relevance
> We agree that the attention has shifted towards Diffusion models but we know that GANs are still widely used in Industry and academia. For example [1] is a very popular recent GAN based model. You can also see [these GAN related papers](https://eccv.ecva.net/virtual/2024/papers.html?filter=titles&search=gan) that were published in ECCV24.
>
> ## Other GAN variants
> We indeed use the linearity of the expectation inside the definition of the Wasserstein distance in our proof. While the same can be done for all IPMs like Sobolev GANs and MMD GANs it may not be directly applied to some other losses like the original or non-saturating GAN losses. However, our experiments with Non-saturating GANs show similar results (See point 4 in our response to reviewer pwLS wher we refer to [this figure](https://postimg.cc/Sn0ZFRWR) (or see appendix F.2 in the updated version).
>
> ## Summary
> Overall we agree with your comment that the paper would benefit a lot from discussing the context of our result and how it relates to other forms of GANs and we will do so in the final version.
>
> [1] Pan, Xingang, et al. "Drag your gan: Interactive point-based manipulation on the generative image manifold." ACM SIGGRAPH 2023 Conference Proceedings. 2023.

---

> > ### Comment · Reviewer_bEHd · 2024-11-27
> >
> > I thank the authors for their detailed answers. I think it would be good to include in the final version a discussion about all other GANs for which the claim "convolutional discriminator -> minimizing patch distributions" holds (for example, all f-divergence GANs).
> >
> > I maintain my initial score.

---

### Official Review · Reviewer_naBZ · 2024-10-30

**Soundness:** 1
**Presentation:** 1
**Contribution:** 1
**Rating:** 1
**Confidence:** 5

**Summary:**

The paper tries to prove that when the discriminator is convolutional, WGANs minimize the Wasserstein distance between patches in the generated images and the training images, not the Wasserstein distance between images. Yet no solid proof is provided.

**Strengths:**

The motivation seems to be good.

**Weaknesses:**

- The symbols used in the paper is chaos, what's $x$, is it the input of the generator or its output?
- The paper is generally not well organized and hard to follow.
- The proofs are not rigorously proven. For example, in theory 3.2, how to justify the so called $N/M$? It is wrong, for example, $M=3, N=5$, while one $x_i$ and $y_j$ overlaps to each other. To prove the theorem, I think the discrete metric need to be introduced.

**Questions:**

- Please see above.
- The paper seems unfinished since there are empty sections in the appendix.

---

> ### Author Response · Authors · 2024-11-19
> **Answer to reviewer naBZ**
>
> Thank you for the time you took for reviewing our work.
>
> - As far as we can tell, x, is only used in section 2 and it denotes samples from the distribution P. Can you please specify what confused you?
> - We didn’t fully understand the reviewer’s question. The discrete W1 metric was introduced in Definition 2.2.
> - We did not have theorem 3.2 in the paper.
> - Thank you for bringing this into our attention. Sections A.6-A.8 refer to figures 9-11 in the appendix. They appear empty due to a formatting error and we will fix this in the final version.

---

### Official Review · Reviewer_pwLS · 2024-11-02

**Soundness:** 3
**Presentation:** 3
**Contribution:** 3
**Rating:** 6
**Confidence:** 3

**Summary:**

This paper presents theoretical and experimental results demonstrating that WGAN actually minimizes the Wasserstein distance (W1), though this depends on the architecture of the discriminator. Using a convolutional discriminator as an example, the paper shows that the W1 distance is minimized over batches rather than individual images.

Specifically, the authors begin with a discrete GAN, where the latent vector z is uniformly sampled from M fixed noise vectors, using this setup as a tool to investigate the Wasserstein distance, as it allows for a more exact computation of the W1 distance. They introduce an iterative algorithm, OTmeans, to compute the W1 distance, serving as a baseline for investigating W1 distance minimization in WGAN. Using this tool, the authors present two main findings in the paper:

Finding 1: Using a 2D discrete GAN to motivate the theorem, the authors observe that when M≥N (where M is the number of noise samples and N is the number of training samples), the discrete GAN reproduces the training examples. Otherwise, it generates outputs as linear combinations of the training examples. They also study cases with images where M=64<N=1000, resulting in blurry images that look similar between WGAN and the OTMeans algorithm. When M=N, there are copied training example.

Finding 2: The authors design two discriminator architectures to demonstrate that optimizes the W1 distance over batches with the convolutional discriminator. The first architecture is a standard convolutional model with convolutional layers followed by a fully connected (FC) layer that operates on batches. The second architecture inserts Global Average Pooling (GAP) between the convolutional layers and the FC layer, making it act on entire images. This setup is then compared with the OTMeans approach on images, where OTMeans uses SGD and Sliced Wasserstein Distance (SWD) to make computations tractable. The authors show that the CNN with GAP behaves similarly to global patch W1, while the other approach resembles local patch W1 distances. They also provide evidence from histograms that demonstrate how GANs learn local patch statistics similar to those in the training set.

**Strengths:**

This is an important finding that sheds light on how WGANs learn, making a valuable contribution to the GAN research community. Overall, it is a well-written paper with interesting results.

**Weaknesses:**

The experiments are limited to simple CNN architectures, and there is no exploration of regularization techniques to enforce the Lipschitz constraint, which might behave differently depending on the hyperparameters used.

**Questions:**

I have a few questions and also some suggestions for further experimenents:

1. Regarding the case where M=N, do all generated examples match the data examples, or are only a few of them exact matches by chance? And what GANs might behave when M>N or M>>N, e.g., do it generate by copying one training sample for more than different fixed noise inputs? Could the authors please provide quantitative results on the percentage of exact matches when M=N and test and report results for cases where M>N and M>>N?

2. WGAN’s learning behavior might also depend on the ratio of M and N. The GANs bahavior could be smoother than just for the concrete cases of M<N, N=N and M>N. Do the authors have any deeper analysis on this aspect? One way to investigate e.g., could the authors conduct experiments with a range of M/N ratios and plot key metrics (e.g., FID score, exact match percentage) as a function of this ratio to visualize if any smooth transitions in behavior.

3. The study is based on a small CNN with two configurations: CNN-GAP and CNN-FC. Is it not clear what method is used to regularize the Lipschitz constraint while training these models. The results might vary considerably depending on the regularization strength and hyperparameters used. If strong regularization were applied, the results could be quite different. Could the authors explicitly provide details of the Lipschitz constraint regularization method used, also the hyperparameters, and conduct an ablation study showing how different regularization strengths affect the results.

4. The study focuses on the W1 distance but DCGAN used in some studies, which also shows similar patch-local distribution behavior. Does this suggest that GANs learn on batches when the discriminator is convolutional, regardless of the GAN loss function used? I’m curious why the authors didn’t use WGAN-GP, given that the study is about the W1 distance. Could the authors include experiments with WGAN-GP for direct comparison, and if possible extending the analysis to other GAN variants to test the generality of the findings across different loss functions?

5. What are Direct_Patch_SWD vs. Direct_LocalPatch_SWD? Can the authors explain how SWD is computed for Direct_Patch_SWD vs. Direct_LocalPatch_SWD?

6. Can the authors explain how to histograms on patches in experiments are generated?

7. Are the generators identical across the two convolutional designs? What was the architecture of the generator used? Could the authors include the details of architectures used in the studies?

---

> ### Author Response · Authors · 2024-11-19
> **Answer to reviewer pwLS**
>
> Thank you for your constructive feedback. We appreciate your comments and suggestions, which will help improve our work.
>
>
> - **Copying with M=N and M>N:** Following the analysis in our paper the optimal solution is to copy all the training set when M=N and to make several copies when M>N (and M is a multiple of N). When we run OT-means, we find that this indeed what happens and all the generated images are copies (similar to the toy data shown in figure 2).  When M>N and is not a multiple of N, then almost all generated images are copies and a small number are averages (e.g. if M=100, N=3, then 99 generated images will be copies and 1 will be an average). In the experiments with WGAN the results depend on convergence parameters but almost all the generated images are clearly noisy copies of some data point. [See here](https://postimg.cc/dhSzywdK) (or see appendix F.4 in the updated version) a larger random batch from the same WGAN from figure 6 in the paper trained on FFHQ.
>
>
> - **Smooth changes with increasing M.** As shown by our theorem, as we increase M gradually each generated image becomes an average of less training examples and the images become gradually sharper. For example, if you have 100 training images and use M=1, then you will have one generated image that is the average of all 100 images, while with M=10, each generated image is an average of 10 training images. As we wrote above, this is easier to see with OT-means because with the WGAN training there may be convergence issues. We will include some results in the final version.
>
> - **Lipschitz constraint regularization:** We indeed forgot to specify that we used Gradient penalty (lambda=10) for all our experiments. Our experiments with different methods for enforcing the Lipschitz constraint (gradient clipping and spectral normalization) did not change the big picture of our findings.
> We will provide details about the method we used in out final revision.
>
> - **WGAN-GP and other GAN variants:** First, There seems to be some misunderstanding of the terms we used and we apologize for not being clearer. All the GAN results in this paper are with WGAN loss and Gradient Penalty. When we refer to DCGAN we mean that the architecture of the discriminator is the one used in the DCGAN paper [1] but we are still using WGAN-GP loss (this was also the case in the WGAN-GP paper [2]). We will clarify this in the next version.
> More to the point, we have found, similar to [3], that using regular GAN loss GAN-NS (non-saturating) with a gradient penalty gives rather similar results to using a WGAN: the GAN-NS approximately minimizes the appropriate W1 (but not as well as a WGAN). [Here is a figure](https://postimg.cc/Sn0ZFRWR) (or see appendix F.2 in the updated version) that compares the results of the experiment from figure 5 in our paper when we change the loss from WGAN (left column) to GAN-NS Loss (middle column)
>
>
>
> - **What is direct_patch_SWD and direct_local_patch_SWD:** In both cases we compute the Sliced Wasserstein Distance (SWD) between sets of patches. In the “local” version we compare patches in the training set and generated set at a single location, while in “direct_patch_SWD” we compare all patches in the two sets of images (disregarding location). As we explain in lines 422-426 the “direct_patch_SWD” results are obtained by using patch SWD as a loss function for the same generator used by the GAN.
>
>
> - **Histograms:** We project all patches in an image with the same random projection into 1d and then plot the histogram of the projections. This is a visualization method to determine whether the patch distributions are the same (if the fake and real images have the same patch distribution then the histograms should align). We describe these in lines 485-486 of the paper and will make sure to write this more clearly.
>
> - **Generator architecture:** This is another important detail we forgot to add. In all our experiments we used the same  FC generator (Appendix F.1 in the new revision shows similar results with convolutional generators). We will add this clarification to our final revision.
>
> Thank you again for your valuable feedback. We look forward to incorporating these improvements.
>
> - [1] Radford, Alec. "Unsupervised representation learning with deep convolutional generative adversarial networks." arXiv preprint arXiv:1511.06434 (2015).
> - [2] Martin Arjovsky, Soumith Chintala, Léon Bottou Proceedings of the 34th International Conference on Machine Learning, PMLR 70:214-223, 2017.
> - [3] William Fedus, Mihaela Rosca, Balaji Lakshminarayanan, Andrew M Dai, Shakir Mohamed, and Ian Goodfellow. Many paths to equilibrium: Gans do not need to decrease a divergence at every step. In International Conference on Learning Representations, 2018.

---

### Author Response · Authors · 2024-11-27
**Updated appendix**

Thank you all for taking the time to review out work and answer our comments.
We have reorganized the appendix and added an additional appendix section, **F**, where we put aditional figures and experiments that were asked for by some of the reviewers.

---

### Meta-Review · Area_Chair_bQnw · 2024-12-22

**Metareview:**

The paper provides new understanding about how Wasserstein GANs (WGANs) work by studying them in the specific setting of discrete generator distributions (i.e., where the noise of the generator is sampled from a discrete set). For this setting, one can directly optimize the Wasserstein distance between the training data and the generated distribution; therefore, this (local) optimal value can be used as a measure to assess how well a WGAN (obtained with e.g. adversarial training) approximates the training data.

This leads to interesting findings. If the number of noise vectors exceeds the number of training data points, then the discrete GAN tends to copy the training data. If the opposite holds, then the discrete GAN copies some, and averages out the rest. (This is supported both by theory and practice). Moreover, the inductive bias of the discriminator is crucial; if the discriminator is convolutional, then the WGAN minimizes the Wasserstein-1 distance between patches of the generated images and patches of the training data. (This is also supported by both theory and practice).

Overall, the paper is of high quality, and makes non-trivial advances in understanding the learning dynamics of Wasserstein GANs (and possibly other related GAN families). I recommend acceptance.

**Additional Comments On Reviewer Discussion:**

Three out of the four reviewers were quite positive about this paper. (The outlier reviewer gave a very low score, bringing the average down, but mainly listed presentation-related complaints.) Reviewers thought that the idea was clever, the paper was well-written, and that the findings matched what has been observed in practice. The authors were able to satisfactorily respond to most points raised by the reviewers.

Some concerns related to practical impact still persisted; for example, do the findings generalize to other losses (such as f-divergences) ? Can anything be said about more practical architectures or regularization schemes? Is the topic timely? While these are important pending questions, I believe that the essence of the paper is very interesting and merits publication.

---

### Decision · Program_Chairs · 2025-01-22

Accept (Poster)